# Resilience and Social Support Protect Nurses from Anxiety and Depressive Symptoms: Evidence from a Cross-Sectional Study in the Post-COVID-19 Era

**DOI:** 10.3390/healthcare13060582

**Published:** 2025-03-07

**Authors:** Aglaia Katsiroumpa, Ioannis Moisoglou, Ioanna V. Papathanasiou, Maria Malliarou, Pavlos Sarafis, Parisis Gallos, Olympia Konstantakopoulou, Fotios Rizos, Petros Galanis

**Affiliations:** 1Clinical Epidemiology Laboratory, Faculty of Nursing, National and Kapodistrian University of Athens, 11527 Athens, Greece; aglaiakat@nurs.uoa.gr (A.K.); olykonstant@nurs.uoa.gr (O.K.); pegalan@nurs.uoa.gr (P.G.); 2Department of Nursing, University of Thessaly, 41500 Larissa, Greece; iopapathanasiou@uth.gr (I.V.P.); malliarou@uth.gr (M.M.); psarafis@uth.gr (P.S.); 3Faculty of Nursing, University of West Attica, 12243 Athens, Greece; parisgallos@nurs.uoa.gr; 4Department of Business Administration, University of West Attica, 12241 Athens, Greece; frizos@uniwa.gr

**Keywords:** resilience, social support, anxiety, depression, nurses

## Abstract

**Background**: Nurses experience high levels of anxiety and depression since they work in a highly stressful environment. Thus, the identification of preventive factors against nurses’ anxiety and depression is essential to improve their quality of life. In this context, our aim was to examine the impact of resilience and social support on nurses’ anxiety and depressive symptoms. **Methods**: A cross-sectional online study was implemented in Greece in September 2024. We used the Brief Resilience Scale, Multidimensional Scale of Perceived Social Support, and Patient Health Questionnaire-4 to measure resilience, social support, anxiety, and depressive symptoms, respectively. **Results**: Our sample included 677 nurses with a mean age of 37.73 years (standard deviation: 9.38). Our multivariable linear regression models identified a negative relationship between resilience and anxiety (adjusted standardized beta coefficient =−0.38; *p* < 0.001) and depressive symptoms (adjusted standardized beta coefficient = −0.36; *p* < 0.001). Similarly, we found that significant others’ support was associated with reduced anxiety (adjusted standardized beta coefficient = −0.27; *p* < 0.001) and depressive symptoms (adjusted standardized beta coefficient = −0.23; *p* < 0.001). The standardized beta coefficient indicated that resilience has a greater impact on anxiety and depressive symptoms than significant others’ support. **Conclusions**: Our findings suggest resilience and social support have a protective function against nurses’ anxiety and depressive symptoms. Managers and policymakers should adopt appropriate interventions to improve nurses’ resilience and social support and, thus, to improve their mental health and quality of life.

## 1. Introduction

Healthcare systems rely heavily on nurses, who provide crucial care and assistance to patients and their families [1,2]. However, the nursing profession is fraught with factors that can negatively affect mental well-being, including high-stress environments, excessive workloads, extended shifts, irregular sleep patterns, exposure to traumatic events, emotional demands, and insufficient support [3,4,5,6]. Consequently, nurses often experience elevated levels of anxiety, depressive symptoms, stress, trauma, and burnout. The consequences of nurses with depression and anxiety are far-reaching, affecting both individuals and organizations. These conditions can compromise the quality and safety of patient care, elevate the likelihood of errors and adverse events, diminish patient satisfaction, and contribute to increased staff turnover and absenteeism [7,8,9,10].

The COVID-19 pandemic has further exacerbated these challenges, presenting nurses with numerous obstacles that continue to affect their work quality and mental well-being. For example, global nurse staffing shortages, intensified by pandemic-related sick leaves due to SARS-CoV-2 infections, have significantly worsened the pre-existing nursing personnel deficit. This has resulted in increased workloads for active nurses [11,12,13]. Nurses during the pandemic experienced high levels of burnout and various mental health issues [14,15]. In particular, two recent meta-analyses, including studies that took place during the COVID-19 pandemic, showed high levels of anxiety and depressive symptoms among nurses [16,17]. Al Maqbali et al. identified 93 studies, including 92,112 nurses, and found that the overall prevalence of anxiety and depressive symptoms was 37% and 35%, respectively [16]. Also, Ślusarska et al. identified 23 studies from nine countries and found that the overall prevalence of anxiety and depressive symptoms among 44,165 nurses was 29% and 22%, respectively [17]. Similarly, an umbrella review of seven meta-analyses found that the overall prevalence of anxiety and depressive symptoms among healthcare workers was 24.9% and 24.8%, respectively [18]. Moreover, burnout, anxiety, depression, and stress were identified as significant mental health concerns during the COVID-19 epidemic in a comprehensive survey of nurses and midwives in Australia [19]. A study conducted in the post-COVID-19 era revealed that mental health issues persist among nursing personnel [20].

Resilience is typically described as the capacity to handle challenging life events, including stress, trauma, threats, and tragedy [21]. It involves the ability to bounce back swiftly from adversity and difficult circumstances [22]. Experts view resilience as a dynamic, evolving psychosocial process through which individuals facing prolonged hardship or potentially harmful experiences achieve positive psychological adaptation over time. This quality is a protective factor for mental and physical well-being, enhancing one’s ability to effectively manage adversity and stressful situations. Rather than being solely an individual trait, resilience involves an interaction between internal and external environmental elements [23]. The literature has demonstrated that people with higher levels of resilience are better equipped to shield themselves from psychiatric disorders [24,25,26]. Numerous early studies have highlighted the preventative effect of resilience on symptoms of anxiety and depressive symptoms across various groups, including cardiovascular disease patients, healthcare professionals, clinical nurses, university students, and pregnant women [25,27,28,29,30]. We should also recognize that several factors related to nurses’ work environment affect their resilience. In particular, the literature suggests that increased levels of work engagement [31,32,33], perceived organizational support [34,35], and innovation-oriented nursing organizational culture [36] improve nurses’ resilience. Moreover, nurses who experience lower levels of workplace violence show higher levels of resilience [37].

The concept of social support encompasses the presence or accessibility of individuals on whom one can depend and from whom one can experience affection, nurturing, and appreciation. In essence, it refers to the actual or perceived external resources an individual can access through their network of friends, family members, romantic partners, and colleagues [38,39]. Social support manifests in various forms, including structural, functional, emotional, instrumental/material, and informational types [40]. Social support plays a crucial role in safeguarding mental well-being. For example, it serves as a key protective factor against depressive symptoms by fostering positive social connections and indirectly shielding individuals from stress [41,42]. Furthermore, high levels of social support have been shown to have a protective effect against post-traumatic stress disorder and burnout [43,44]. We should notice that leadership promoting a better work environment for nurses may contribute to social support for them [45]. Moreover, organizational support may improve nurses’ social support [46]. Several other factors, such as occupational commitment [47], sense of coherence [48,49], and positive strategies (i.e., occupational coping self-efficacy) [50], may contribute to nurses’ social support.

The importance of resilience and social support in nurses’ mental health emerged during the COVID-19 pandemic. In particular, several studies during the COVID-19 pandemic showed that resilience plays a protective role against anxiety [51,52,53] and depressive symptoms [25,51,53,54,55]. However, all these studies were conducted only in Asian countries, such as China [52], South Korea [54,55], Iran [51], Taiwan [53], and Turkey [25]. Similarly, five other studies in Asian countries (four in China and one in Japan) showed a negative association between social support, anxiety, and depressive symptoms [44,56,57,58,59]. However, after the end of the COVID-19 pandemic, no further studies were carried out to investigate the association between resilience, social support, anxiety, and depressive symptoms. Additionally, the literature on the impact of resilience and social support on nurses’ anxiety and depressive symptoms in Europe is very limited. In this context, our aim was to investigate the impact of resilience and social support on anxiety and depressive symptoms in a sample of nurses in Greece. In short, our research hypotheses included the following:-Resilience reduces anxiety in nurses.-Social support reduces anxiety in nurses.-Resilience reduces depressive symptoms in nurses.-Social support reduces depressive symptoms in nurses.

## 2. Materials and Methods

### 2.1. Study Design

A cross-sectional online study was implemented in Greece in September 2024. We applied the Strengthening the Reporting of Observational Studies in Epidemiology (STROBE) guidelines [60]. We used Google Forms to create a digital version of the study questionnaire, which was subsequently distributed through Facebook groups for nurses. Thus, we obtained a convenience sample. Participants were required to meet the following criteria: (1) be a practicing nurse in healthcare facilities such as hospitals, health centers, or nursing homes; (2) have a minimum of one year of work experience; and (3) provide written informed consent for study participation. No compensation was offered to those who took part in the study.

We used G*Power v.3.1.9.2 to calculate our sample size. In our study, we considered two predictors (resilience and social support) and five confounders (sex, age, educational level, work experience, and self-assessment of health status). Considering an anticipated effect size of 0.02 between resilience (or social support) and outcomes (anxiety and depressive symptoms), a statistical power of 95%, and a margin of error of 5%, the sample size was estimated to be 652 nurses.

### 2.2. Measurements

We measured several demographic characteristics, including sex, age, MSc/PhD diploma, work experience, and self-assessment of health status.

We used the Brief Resilience Scale (BRS), which includes six items [61] to measure resilience. Answers utilize a five-point Likert scale that ranges from 1 (strongly disagree) to 5 (strongly agree). The score on the BRS is calculated as an average of all answers; thus, the total score ranges from 1 to 5. Higher scores indicate a higher level of resilience. We used the valid Greek version of the BRS [62]. In our study, Cronbach’s alpha for the BRS was 0.804.

Social support was measured with the Multidimensional Scale of Perceived Social Support (MSPSS) [63]. The Greek version of the MSPSS has been translated and validated [64]. The MSPSS includes 12 items and measures three dimensions of social support: family support (e.g., “My family really tries to help me”); friends’ support (e.g., “I can count on my friends when things go wrong”); significant others’ support (e.g., “There is a special person who is around when I am in need”). Answers utilize a seven-point Likert scale that ranges from 1 (very strongly disagree) to 7 (very strongly agree), with higher scores indicative of higher levels of support. We found that Cronbach’s alpha for the three factors was greater than 0.922. In particular, Cronbach’s alpha for the “family support” factor was 0.952, for the “friends’ support” factor, it was 0.954, and for the “significant others’ support” factor, it was 0.922.

Anxiety and depressive symptoms were measured with the Patient Health Questionnaire-4 (PHQ-4) [65]. Answers utilize a four-point Likert scale that ranges from 0 (not at all) to 3 (nearly every day), with higher scores indicative of higher levels of anxiety and depressive symptoms. Experts should further examine participants with a score ≥3 in each factor. The PHQ-4 has been translated and validated in Greek [66]. We found that Cronbach’s alpha for the factors “anxiety” and “depressive symptoms” was 0.827 and 0.769, respectively.

### 2.3. Ethical Issues

Our study protocol received approval from the Faculty of Nursing Ethics Committee at the National and Kapodistrian University of Athens (approval number 476; approved: November 2023). Moreover, our study adhered to the principles of the Declaration of Helsinki [67]. On the first page of the questionnaire, the study’s purpose and data use were explained. Participants were also informed that the survey should take approximately 8 min to complete, whilst their consent was granted upon its completion. The survey’s completion was voluntary and anonymous. Participants had the right to withdraw from the study at any time during data collection. We did not collect personal data from nurses.

### 2.4. Statistical Analysis

We used numbers and percentages to present categorical variables. Moreover, we used mean, standard deviation (SD), median, minimum, and maximum values to present continuous variables. The Kolmogorov–Smirnov test and Q-Q plots showed that anxiety and depressive symptoms scores follow normal distribution. First, we performed a univariate linear regression analysis between independent variables and anxiety and depressive symptoms. Then, we constructed two multivariable linear regression models with anxiety and depressive symptoms as the dependent variables. We adjusted our multivariable models for sex, age, educational level, work experience, and self-assessment of health status. We calculated unadjusted and adjusted beta coefficients, 95% confidence intervals (CIs), and *p*-values. *P*-values less than 0.05 were considered statistically significant. We assessed multicollinearity issues by calculating variance inflation factors (VIFs). VIFs greater than 5 indicate multicollinearity between independent variables [68]. We examined the histograms of the residuals to check multivariable normality. We examined scatterplots of residuals versus predicted values to check for homoscedasticity and linearity. We used Harman’s single factor to detect common method bias. Common method bias occurs when the unrotated solution produces one factor that accounts for more than 50% of the variance [69,70]. We did not find common method bias in our study since our unrotated solution produced one factor that accounted for 38.1% of the variance. We used IBM SPSS 21.0 (IBM Corp; released 2012; IBM SPSS Statistics for Windows, Version 21.0. Armonk, NY, USA: IBM Corp) for the analysis.

## 3. Results

### 3.1. Demographic Characteristics

Our sample consisted of 677 nurses with a mean age of 37.73 years. Among our nurses, 89.4% were females, and 54.4% possessed a MSc/PhD diploma. Among our nurses, 89.7% considered their health status as good/very good. The mean work experience was 11.59 years. Demographic characteristics of nurses are shown in Table 1.

### 3.2. Study Scales

The mean resilience score was 3.45 (SD: 0.65), indicating a moderate level of resilience among our nurses. Levels of social support were high in our sample. In particular, we found that our nurses received higher levels of significant others’ support (mean: 6.02; SD: 1.46) and family support (mean: 5.94; SD: 1.51) but lower levels of friends’ support (mean: 5.77; SD: 1.47). The mean anxiety score was 2.35 (SD: 1.62), while the mean depressive symptoms score was 2.15 (SD: 1.58). Among our nurses, 38.0% (n = 257) had an anxiety score of 3 or greater, indicating the presence of anxiety symptoms. Moreover, 32.6% (n = 221) of our nurses had depressive symptoms score ≥ 3, which was indicative of depressive symptoms. Table 2 shows descriptive statistics for the study scales, while Table 3 shows correlations between study scales.

### 3.3. Impact of Resilience and Social Support on Anxiety and Depressive Symptoms

Our multivariable linear regression analysis showed that resilience and social support reduce nurses’ anxiety (Table 4). After adjustment for sex, age, educational level, work experience, and self-assessment of health status, we found a negative relationship between resilience and anxiety (adjusted standardized beta coefficient = −0.38; *p* < 0.001). Additionally, we found that significant others’ support is an independent preventive factor of anxiety (adjusted standardized beta coefficient = −0.27; *p* < 0.001). Considering the standardized beta coefficient, the impact of resilience on anxiety was higher than that of significant others’ support. The other two dimensions of social support (family and friends’ support) were statistically significant predictors of anxiety in the univariate models but were not in the final multivariable model. There were no multicollinearity issues since VIFs for independent variables were lower than the acceptable value of 5. Figure 1 indicates multivariable normality since the residuals followed the normal distribution. Figure 2 indicates the homoscedasticity and linearity of the multivariable model with anxiety as the dependent variable. We discovered similar findings regarding the impact of social support on anxiety and depressive symptoms. In particular, our multivariable linear regression model identified a negative relationship between resilience (adjusted standardized beta coefficient = −0.36; *p* < 0.001) and significant others’ support (adjusted standardized beta coefficient = −0.23; *p* < 0.001) and depressive symptoms. The standardized beta coefficient indicated that resilience has a greater impact on depressive symptoms than significant others’ support. Family and friends’ support were significant predictors of depressive symptoms in the univariate linear regression models, but this relationship was eliminated after the multivariable analysis. Table 5 shows linear regression models with depressive symptoms as the dependent variable. There were no multicollinearity issues since VIFs for independent variables were lower than the acceptable value of 5. Figure 3 indicates multivariable normality since the residuals followed the normal distribution. Figure 4 indicates the homoscedasticity and linearity of the multivariable model with depressive symptoms as the dependent variable.

## 4. Discussion

To the best of our knowledge, this is the first study that examines the association between resilience, social support, anxiety, and depressive symptoms after the COVID-19 pandemic. Moreover, for the first time, we investigated the association between these variables in a European country and not only after the pandemic. We found that resilience and social support play a protective role against nurses’ anxiety and depressive symptoms. Moreover, our findings indicated that a significant percentage of nurses experience anxiety and depressive symptoms.

In particular, scores on the PHQ-4 indicated that 38.0% of our nurses experienced anxiety symptoms, while 32.6% experienced depressive symptoms. The literature supports these findings since a meta-analysis, including studies during the COVID-19 pandemic, found that the overall prevalence of nurses’ anxiety and depressive symptoms is 37% and 35%, respectively [16]. Also, a similar meta-analysis found a slightly lower overall prevalence of anxiety and depressive symptoms in nurses during the pandemic: 29% and 22%, respectively [17]. Additionally, an umbrella review of seven meta-analyses found that the overall prevalence of anxiety and depressive symptoms among healthcare workers during the pandemic was 24.9% and 24.8%, respectively [18]. It is, therefore, clear that the high levels of anxiety and depressive symptoms experienced by nurses remain after the end of the pandemic. Indicatively, in studies conducted after the pandemic, the prevalence of anxiety ranged from 28.7% to 39.6%, while the prevalence of depressive symptoms ranged from 13.6% to 52.1% [71,72,73,74]. Moreover, our findings were confirmed by a study in Greece including 380 nurses after the pandemic that found that 33.3% of nurses experience anxiety symptoms and 35.0% experience depressive symptoms [75]. The elevated prevalence of anxiety and depressive symptoms among nurses can be attributed to their constant exposure to various stressors in their daily work environment. The COVID-19 pandemic has further exacerbated the already challenging working conditions for nurses. For example, many countries are now grappling with a widespread shortage of nursing staff. Furthermore, nurses’ mental well-being is jeopardized by numerous factors, including overwhelming workloads, prolonged work hours, emotionally taxing situations, encounters with traumatic events, and disrupted sleep schedules [4,5,6,11,12,13].

Our findings indicate that resilience plays a protective role against nurses’ anxiety and depressive symptoms. We found that the nurses in our sample experience moderate levels of resilience. Two studies in Greece confirmed these moderate levels of resilience, including those of nurses working after the COVID-19 pandemic [76,77]. In particular, the mean resilience score in our study was 3.50, while in these two studies, it ranged between 3.40 and 3.43. Several studies during the COVID-19 pandemic confirmed that resilience is associated with reduced anxiety and depressive symptoms among nurses [25,51,52,53,54,55]. In other words, nurses with higher resilience levels have fewer anxiety and depressive symptoms. Nurses who possess high levels of resilience are better equipped to combat stressors and mitigate their detrimental psychological effects. Additionally, resilience encourages nurses to accept and take on responsibilities, reducing burnout and improving their physical and mental well-being [78,79,80,81]. Consequently, resilience plays a crucial role in minimizing the negative impact of work-related stress and enhancing nurses’ ability to adapt and thrive in challenging healthcare environments [82]. Resilience aids in diminishing and overcoming negative emotions such as anxiety and depression. On the other hand, nurses with low resilience struggle to effectively manage or control stress. Resilience helps to counteract or modify the adverse effects of unfavorable workplace conditions, enhances the psychological health of nursing staff, and improves the quality of care provided by nurses. Without resilience, nurses may experience poor working conditions, which can lead to psychological harm [83,84,85,86,87]. The resilience of nurses is essential for their mental well-being and the quality of care; thus, fostering and augmenting resilience should be a primary responsibility of healthcare organizations. Elements that have demonstrated a beneficial impact on the cultivation of nursing resilience encompass formal educational initiatives, social support, and meaningful recognition [88]. Specifically, supervisory social support encompasses strategies such as fostering teamwork, conducting debriefing and reflection, establishing social connections, encouraging nurses to adopt a positive perspective on events, and leveraging nurses’ abilities [89,90]. Social support, therefore, and, in particular, significant others’ support (such as that of the supervisor), can directly affect nurses’ mental health, according to the findings of the present study, and has an indirect effect through prior resilience enhancement.

Our multivariable analysis identified significant others’ support as an independent predictor of anxiety and depressive symptoms. In particular, we found that nurses receiving higher levels of significant others’ support have fewer anxiety and depressive symptoms. Similar studies in China and Japan confirm our findings since they found that social support reduces nurses’ anxiety and depressive symptoms [44,56,57,58,59]. Significant others’ support may include colleagues and/or supervisors for employees. Since the nursing profession is highly demanding and faces many challenges in the work environment, nurses may consider that support from colleagues and/or supervisors is crucial. This is a possible explanation for our findings that significant others’ support was more important than family and friends’ support among our nurses. We should also recognize that maybe mediators or confounders exist in the relationship between social support, anxiety, and depressive symptoms that we did not measure in our study. For instance, personality characteristics and emotional intelligence may affect this relationship. External resources that individuals perceive or have access to from their personal connections, including friends, family members, romantic partners, and colleagues, are social support. This support plays a crucial role in mental health, particularly for nurses, as it allows them to express their stressors and helps prevent or manage anxiety and depressive symptoms [91]. Research has shown that social support is an effective tool for coping with stress, and individuals who receive such assistance are better equipped to handle the negative effects of stress [92]. Additionally, social support protects against anxiety and depressive symptoms by fostering positive social relationships and indirectly shielding individuals from stress [41,42,87,93,94]. Furthermore, social support can be described as a network of psychological and material resources available to individuals as they cope with anxiety and depressive symptoms [40]. In this context, nurses receive varying degrees of support from their family, friends, and significant others, which enables them to comprehend and value the challenges they face in balancing their work and personal life demands [56].

### Limitations

Our study had several limitations. First, we employed a cross-sectional design to examine the association between resilience, social support, anxiety, and depressive symptoms. Therefore, we cannot establish causal relationships between these variables. Longitudinal studies are necessary to identify the association between resilience, social support, anxiety, and depressive symptoms over time. Second, we obtained a convenience sample through a web-based survey. For instance, our nurses may be younger than those in the source population of nurses in Greece since younger individuals are more active on social media. Moreover, in our study, there was an over-representation of females (89.4%) since recent statistics showed that the percentage of female Facebook users in 2024 was 43% [95]. Thus, our sample was not representative of all nurses in Greece. Although we achieved the minimum required sample size, we cannot generalize our findings. Future studies with random and stratified samples of nurses could add significant information. Third, we performed our study in a European country, and thus, we cannot make inferences about nurses in other countries. Therefore, it is necessary to implement further studies worldwide to confirm our findings. Fourth, although we used valid tools to measure resilience, social support, anxiety, and depressive symptoms among nurses, information bias is probable in our study since these tools are self-reported measurements. Fifth, we constructed our multivariable models by eliminating several confounders. However, several other variables may act as confounders in the association between resilience, social support, anxiety, and depressive symptoms. Future studies should eliminate more confounders to obtain more valid results.

## 5. Conclusions

Our findings suggest that higher levels of resilience and better social support result in less anxiety and depressive symptoms among nurses. Thus, resilience and social support can contribute to preventing and managing nurses’ anxiety and depressive symptoms. Nursing managers and policymakers should focus on identifying key groups for targeted interventions and provide comprehensive training to nurses. This training should aim to enhance their resilience, social support networks, and ability to manage workplace stressors. Research has underscored the positive impact of training programs on augmenting nurses’ resilience, concurrently leading to reductions in stress, depression, anxiety, and burnout levels [96]. Both organizational support and that of the supervisor can enhance nurses’ resilience and reduce the likelihood of anxiety [97]. Additionally, initiatives should be developed to bolster resilience and social support, with particular emphasis on nurses working in high-risk units. It is essential for nurses to cultivate workplace resilience, especially during challenging and critical work situations, enabling them to transform negative experiences into positive ones. Implementing psychological empowerment strategies for nurses in hospital settings may contribute to reducing anxiety and depressive symptoms while improving their resilience skills. By focusing on strengthening resilience and offering robust social support to nurses, healthcare organizations may effectively reduce the prevalence of anxiety and depressive symptoms among these vital healthcare professionals.

## Figures and Tables

**Figure 1 healthcare-13-00582-f001:**
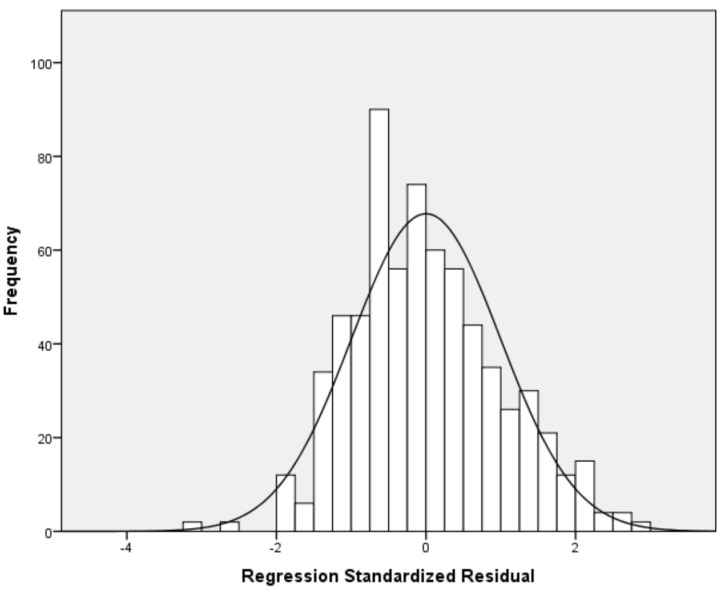
Histogram of the residuals with anxiety as the dependent variable.

**Figure 2 healthcare-13-00582-f002:**
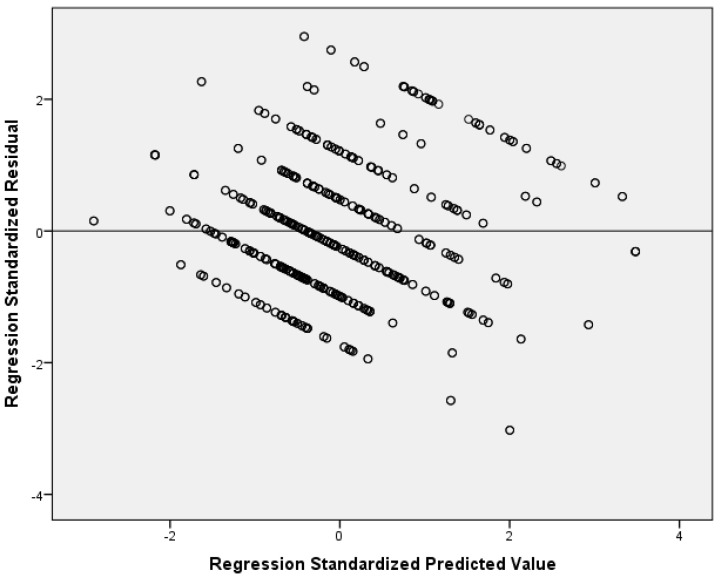
Scatterplot of residuals versus predicted values with anxiety as the dependent variable.

**Figure 3 healthcare-13-00582-f003:**
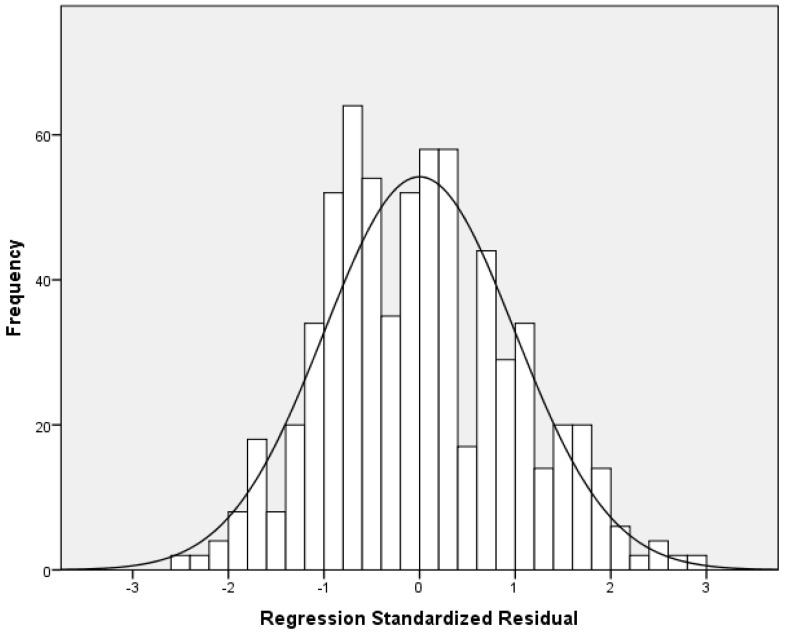
Histogram of the residuals with depressive symptoms as the dependent variable.

**Figure 4 healthcare-13-00582-f004:**
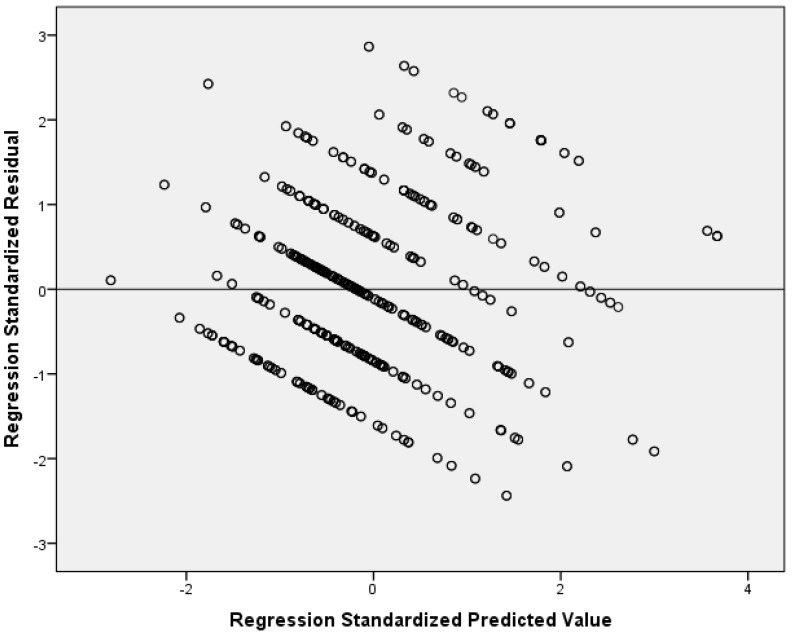
Scatterplot of residuals versus predicted values with depressive symptoms as the dependent variable.

**Table 1 healthcare-13-00582-t001:** Demographic characteristics of nurses (n = 677).

Characteristics	N	%
Sex		
Males	72	10.6
Females	605	89.4
Age (years) ^a^	37.73	9.38
MSc/PhD diploma		
No	309	45.6
Yes	368	54.4
Self-assessment of health status		
Very poor	18	2.7
Poor	10	1.5
Moderate	42	6.2
Good	411	60.7
Very good	196	29.0
Work experience (years) ^a^	11.59	8.95

^a^ mean, standard deviation.

**Table 2 healthcare-13-00582-t002:** Descriptive statistics for the study scales (n = 677).

Scale	Mean	Standard Deviation	Median	Minimum Value	Maximum Value
Resilience	3.45	0.65	3.5	1	5
Family support	5.94	1.51	6.5	1	7
Friends’ support	5.77	1.47	6.3	1	7
Significant others’ support	6.02	1.46	6.5	1	7
Anxiety	2.35	1.62	2.0	0	6
Depressive symptoms	2.15	1.58	2.0	0	6

**Table 3 healthcare-13-00582-t003:** Correlations between study scales (n = 677).

Scale	1	2	3	4	5	6
Resilience		0.23 **	0.21 **	0.26 **	−0.45 **	−0.43 **
2. Family support			0.66 **	0.79 **	−0.28 **	−0.29 **
3. Friends’ support				0.69 **	−0.26 **	−0.27 **
4. Significant others’ support					−0.36 **	−0.35 **
5. Anxiety						0.76 **
6. Depressive symptoms						

** *p*-value < 0.01.

**Table 4 healthcare-13-00582-t004:** Linear regression models with anxiety as the dependent variable (n = 677).

Independent Variables	Univariate Models				Multivariable Model ^a,b^				
	Unadjusted Unstandardized Beta Coefficient	95% CI for Beta	Unadjusted Standardized Beta Coefficient	*p*-Value	Adjusted Unstandardized Beta Coefficient	95% CI for Beta	Adjusted Standardized Beta Coefficient	*p*-Value	VIF
Resilience	−1.12	−1.29 to −0.95	−0.45	<0.001	−0.94	−1.11 to −0.77	−0.38	<0.001	1.145
Family support	−0.30	−0.38 to −0.23	−0.28	<0.001	0.05	−0.07 to 0.17	0.05	0.389	2.849
Friends’ support	−0.29	−0.37 to −0.21	−0.26	<0.001	−0.05	−0.16 to 0.05	−0.05	0.299	2.122
Significant others’ support	−0.39	−0.47 to −0.32	−0.36	<0.001	−0.30	−0.43 to −0.18	−0.27	<0.001	3.090

^a^ Multivariable model is adjusted for sex, age, educational level, work experience, and self-assessment of health status; ^b^ R^2^ for the multivariable model = 29.1%, *p*-value for ANOVA < 0.001; CI: confidence interval; VIF: variance inflation factor.

**Table 5 healthcare-13-00582-t005:** Linear regression models with depressive symptoms as the dependent variable (n = 677).

Independent Variables	Univariate Models				Multivariable Model ^a,b^				
	Unadjusted Unstandardized Beta Coefficient	95% CI for Beta	Unadjusted Standardized Beta Coefficient	*p*-Value	Unadjusted Unstandardized Beta Coefficient	95% CI for Beta	Adjusted Standardized Beta Coefficient	*p*-Value	VIF
Resilience	−1.04	−1.21 to −0.88	−0.43	<0.001	−0.88	−1.05 to −0.71	−0.36	<0.001	1.145
Family support	−0.30	−0.38 to −0.23	−0.29	<0.001	0.001	−0.11 to 0.12	0.001	0.992	2.849
Friends’ support	−0.29	−0.37 to −0.21	−0.27	<0.001	−0.05	−0.15 to 0.05	−0.05	0.321	2.122
Significant others’ support	−0.37	−0.45 to −0.29	−0.35	<0.001	−0.24	−0.37 to −0.12	−0.23	<0.001	3.090

^a^ Multivariable model is adjusted for sex, age, educational level, work experience, and self-assessment of health status; ^b^ R^2^ for the multivariable model = 26.1%; *p*-value for ANOVA < 0.001; CI: confidence interval; VIF: variance inflation factor.

## Data Availability

The original data presented in this study are openly available in FigShare at https://doi.org/10.6084/m9.figshare.27942213 (accessed on 2 December 2024).

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
