# Peer review of "Resilience and Social Support Protect Nurses from Anxiety and Depressive Symptoms: Evidence from a Cross-Sectional Study in the Post-COVID-19 Era"

_healthcare, 2025, doi:10.3390/healthcare13060582_

Round 1

Reviewer 1 Report

Comments and Suggestions for Authors

The manuscript reported the results of the role of resilience and social support in mitigating anxiety and depression among nurses in Greece. The manuscript is well-structured, employs appropriate methodologies, and makes a meaningful contribution to the field. However, certain aspects require further clarification and refinement to enhance the manuscript's clarity, rigor, and interpretability. Below are my specific comments and suggestions:

  1. Title: The title is clear but could be made more precise if it explicitly mentioned that the study is based on a cross-sectional survey.
  1. Abstract: It would be beneficial to report standardized beta (β) coefficients rather than unstandardized beta (b) coefficients. Since resilience and social support were measured on different scales, using unstandardized coefficients makes it challenging to compare the relative importance of each predictor. Standardized beta coefficients, which transform variables into z-scores, facilitate comparison and enhance the generalizability of findings across studies using different measurement units.
  1. Study Design and Sampling: The use of convenience sampling through Facebook groups introduces a potential selection bias, as nurses who are more active on social media may differ systematically from those who are not. This limitation should be more explicitly acknowledged and discussed, particularly in terms of how it may have influenced the sample’s representativeness.
  1. Statistical analysis: While multiple linear regression is appropriately used, the manuscript does not discuss potential multicollinearity issues. Were any variance inflation factors (VIFs) assessed?
  1. Results: In Tables 3 and 4, as well as in the text describing the results, it is recommended to report standardized beta coefficients (β) instead of unstandardized beta coefficients (b). This will improve interpretability and allow for meaningful comparisons across variables measured on different scales.
  1. Major health organizations, such as the American Psychological Association (APA), recommend using sex when referring to biological categories and gender when discussing social identities or roles. Since the study collected biological sex, it would be more appropriate to use "sex" rather than "gender" in the manuscript.
  1. Discussion: (line 264 to 281): The statement "Our multivariable analysis identified social support as an independent predictor of anxiety and depression" should be refined for accuracy. Based on the multiple regression analysis, only "significant others support" was significantly associated with anxiety and depression, whereas "family support" and "friend support" were not statistically significant in the final model.
  1. Discussion: The authors should clarify the distinction between "significant others support" and “social support”, and provide a possible explanation for why family support and friend support became non-significant when significant others support was included in the model. For example, a discussion of potential mediating or confounding effects could strengthen the interpretation of these findings.

Author Response

Comment

  1. Title: The title is clear but could be made more precise if it explicitly mentioned that the study is based on a cross-sectional survey.

Reply: Done

We rewrite the title as follows.

Resilience and social support protect nurses from anxiety and depression: evidence from a cross-sectional study in the post-COVID-19 era

Comment

  1. Abstract: It would be beneficial to report standardized beta (β) coefficients rather than unstandardized beta (b) coefficients. Since resilience and social support were measured on different scales, using unstandardized coefficients makes it challenging to compare the relative importance of each predictor. Standardized beta coefficients, which transform variables into z-scores, facilitate comparison and enhance the generalizability of findings across studies using different measurement units.

Reply: Done

We rewrite abstract as follows.

…Our multivariable linear regression models identified a negative relationship between resilience and anxiety (adjusted standardized coefficient beta=-0.38, p<0.001) and depression (adjusted standardized coefficient beta=-0.36, p<0.001). Similarly, we found that significant others support was associated with reduced anxiety (adjusted standardized coefficient beta=-0.27, p<0.001) and depression (adjusted standardized coefficient beta=-0.23, p<0.001). Standardized coefficient betas indicated that resilience has a greater impact on depression and anxiety than significant others support…

Comment

  1. Study Design and Sampling: The use of convenience sampling through Facebook groups introduces a potential selection bias, as nurses who are more active on social media may differ systematically from those who are not. This limitation should be more explicitly acknowledged and discussed, particularly in terms of how it may have influenced the sample’s representativeness.

Reply: Done

We add the following text in the Limitations section.

For instance, our nurses may be younger than those in source population of nurses in Greece since younger individuals are more active on social media. Moreover, in our study there was an over-representation of females (89.4%) since recent statistics showed that the percentage of female Facebook users in 2024 was 43% [75].

Comment

  1. Statistical analysis: While multiple linear regression is appropriately used, the manuscript does not discuss potential multicollinearity issues. Were any variance inflation factors (VIFs) assessed?

Reply: Done

We add the following text in the Statistical analysis section.

We assessed multicollinearity issues by calculating variance inflation factors (VIFs). VIF greater than 5 indicates multicollinearity between independent variables [53].

Also, we add variance inflation factors in Tables 3 and 4. Please see Tables 3 and 4 in the manuscript.

Moreover, we add the following text in the results.

..There were no multicollinearity issues since VIFs for independent variables were lower than the acceptable value of 5…

Comment

  1. Results: In Tables 3 and 4, as well as in the text describing the results, it is recommended to report standardized beta coefficients (β) instead of unstandardized beta coefficients (b). This will improve interpretability and allow for meaningful comparisons across variables measured on different scales.

Reply: Done

We add standardized beta coefficients in Tables 3 and 4. Please see Tables 3 and 4 in the manuscript.

We rewrite results as following.

…After adjustment for gender, age, educational level, work experience, and self-assessment of health status we found a negative relationship between resilience and anxiety (adjusted standardized coefficient beta=-0.38, p<0.001). Additionally, we found that significant others support is an independent preventive factor of anxiety (adjusted standardized coefficient beta=-0.27, p<0.001). Considering standardized coefficient betas the impact of resilience on anxiety was higher than that of significant others support....

… In particular, our multivariable linear regression model identified a negative relationship between resilience (adjusted standardized coefficient beta=-0.36, p<0.001) and significant others support (adjusted standardized coefficient beta=-0.23, p<0.001), and depression. Standardized coefficient betas indicated that resilience has a greater impact on depression than significant others support. …

Comment

  1. Major health organizations, such as the American Psychological Association (APA), recommend using sex when referring to biological categories and gender when discussing social identities or roles. Since the study collected biological sex, it would be more appropriate to use "sex" rather than "gender" in the manuscript.

Reply: Done

We replace “gender” with “sex”.

Comment

  1. Discussion: (line 264 to 281): The statement "Our multivariable analysis identified social support as an independent predictor of anxiety and depression" should be refined for accuracy. Based on the multiple regression analysis, only "significant others support" was significantly associated with anxiety and depression, whereas "family support" and "friend support" were not statistically significant in the final model.

Reply: Done

We rewrite this sentence as following.

Our multivariable analysis identified significant others support as an independent predictor of anxiety and depression. In particular, we found that nurses receiving higher levels of significant others support have fewer anxiety and depressive symptoms.

Comment

  1. Discussion: The authors should clarify the distinction between "significant others support" and “social support”, and provide a possible explanation for why family support and friend support became non-significant when significant others support was included in the model. For example, a discussion of potential mediating or confounding effects could strengthen the interpretation of these findings.

Reply: Done

We add the following text in the Discussion section.

Significant other support may include colleagues or/and supervisors for employees. Since nursing profession is a highly demand job with many challenges in work environment, nurses may consider that support from colleagues or/and supervisors is crucial. This is a possible explanation for our finding that significant others support was more important than family and friends support among our nurses. We should also recognize that may be exist mediators or confounders in the relationship between social support, anxiety, and depression that we did not measure in our study. For instance, personality characteristics and emotional intelligence may affect this relationship.

Reviewer 2 Report

Comments and Suggestions for Authors

Review of Manuscript Number HEALTHCARE-3494583
"Resilience and Social Support Protect Nurses from Anxiety and Depression: Evidence from Greece in the Post-COVID-19 Era"

I found this manuscript to be an interesting and relevant contribution to the literature on nurses’ mental health. The study utilizes data from 677 nurses in Greece to examine how higher levels of resilience and social support are associated with lower levels of anxiety and depressive symptoms. The findings have important implications for policymaking, particularly in designing interventions to support nurses' well-being in the post-pandemic period.

I provide the following minor comments to enhance the quality of the paper:

  1. Title: The current title is too lengthy. I suggest shortening it to a maximum of 12 words for clarity and conciseness.

  2. Literature Review: There are recent studies assessing the mental well-being of nurses during and after the COVID-19 pandemic. Incorporating an analysis of trends in nurses' mental health based on recent research would strengthen the manuscript and ensure the references are more current. The following studies may be relevant:

    • Zamanzadeh, A., Eckert, M., Corsini, N., Adelson, P., & Sharplin, G. (2025). Mental health of Australian frontline nurses during the COVID-19 pandemic: Results of a large national survey. Health Policy, 151, 105214.
    • Yang, S., Hao, Q., Sun, H., Yang, Y., Liu, J., Li, C., ... & Luo, G. (2025). Prevalence and correlates of severe anxiety among front-line nurses during and after the COVID-19 pandemic: a large-scale multi-center study. BMC Nursing, 24(1), 54.
  3. Conclusion: The conclusion would benefit from including practical recommendations grounded in the findings. Specifically, the authors should provide a more comprehensive policy discussion, referencing other studies that offer policy insights and suggesting future research directions relevant to post-pandemic workforce well-being.

Overall, this is a valuable study with policy significance. Addressing these minor revisions will further strengthen its contribution.

Author Response

Comment

Title: The title is clear but could be made more precise if it explicitly mentioned that the study is based on a cross-sectional survey.

Reply:

Actually, the title is rather long, but this is because the authors' intention was to identify the period of the study (post-COVID-19 era) and the country of implementation. Therefore, we retained the existing title.

Comment

Literature Review: There are recent studies assessing the mental well-being of nurses during and after the COVID-19 pandemic. Incorporating an analysis of trends in nurses' mental health based on recent research would strengthen the manuscript and ensure the references are more current. The following studies may be relevant:

    • Zamanzadeh, A., Eckert, M., Corsini, N., Adelson, P., & Sharplin, G. (2025). Mental health of Australian frontline nurses during the COVID-19 pandemic: Results of a large national survey. Health Policy, 151, 105214.
    • Yang, S., Hao, Q., Sun, H., Yang, Y., Liu, J., Li, C., ... & Luo, G. (2025). Prevalence and correlates of severe anxiety among front-line nurses during and after the COVID-19 pandemic: a large-scale multi-center study. BMC Nursing, 24(1), 54.

Reply: Done

Both studies were incorporated into the Literature Review section.

Comment

Conclusion: The conclusion would benefit from including practical recommendations grounded in the findings. Specifically, the authors should provide a more comprehensive policy discussion, referencing other studies that offer policy insights and suggesting future research directions relevant to post-pandemic workforce well-being.

Reply: Done

We add the following text in the Conclusion section: “Research has underscored the positive impact of training programs on augmenting nurses' resilience, concurrently leading to reductions in stress, depression, anxiety, and burnout levels [96]. Both organisational support and that of the supervisor can enhance nurses' resilience and reduce the likelihood of anxiety [97]”.

Reviewer 3 Report

Comments and Suggestions for Authors

Dear Authors,

I read the manuscript entitled “Resilience and social support protect nurses from anxiety and depression: evidence from Greece in the post-COVID-19 era”.

Nurses experience high levels of anxiety and depression since they work in a high stressful environment. The working environment has even got worse because of the COVID-19 pandemic. In this context, identification of preventive factors against nurses’ anxiety and depression is essential to improve their quality of life and productivity. Therefore, this study is important since it investigates the association between resilience, social support, anxiety and depression. Overall, the quality of the manuscript is high and you provide significant information on this issue in a valid way.

I suggest you to take into consideration the following comments.

In the Introduction section:

You explain the scientific background and rationale for the investigation of the association between resilience, social support, anxiety and depression. Also, you state specific objectives in a clear way.

As resilience is vital for nurses' mental health, the factors that contribute to resilience and which are related to nurses' work environment should be presented. In particular, how supervisor or organizational support for nurses can enhance resilience.

Similarly for social support, what actions of leadership at the unit and organizational level contribute to social support for nurses.

In the Methods section:

You present key elements of study design and describe in detail the setting, locations, and relevant dates, including periods of data collection. You report inclusion and exclusion criteria, and define all outcomes, predictors, and confounders.

For each variable of interest, you give sources of data and details of methods of assessment.

Moreover, you explain the calculation for the sample size, and describe all statistical methods, including those used to control for confounding.

In the Results section:

You report numbers of individuals at each stage of study, and give characteristics of study participants (e.g., demographic,) and information on predictors and potential confounders.

Also, you report unadjusted and adjusted estimates and their 95% confidence interval, and clarify which confounders were adjusted for and why they were included.

In my opinion, in the section on the demographic characteristics of the participants, there is no need for a detailed presentation of the results, as these are presented in Table 1.

In the Discussion section:

You summarise key results with reference to study objectives. Also, you give a cautious overall interpretation of results considering objectives, limitations, and results from similar studies. Moreover, you discuss limitations of the study, taking into account sources of potential bias or imprecision.

Two of the important findings of the study are that significant others support is negatively related to both anxiety and depression. As significant others include colleagues and supervisors, you should highlight their role in reducing nurses' anxiety and depression.

In addition, as you have measured various variables (e.g., resilience, anxiety, depression) in your study, please compare your finding with similar studies in Greece.

Good luck

Author Response

Comment

In the Introduction section:

You explain the scientific background and rationale for the investigation of the association between resilience, social support, anxiety and depression. Also, you state specific objectives in a clear way.

As resilience is vital for nurses' mental health, the factors that contribute to resilience and which are related to nurses' work environment should be presented. In particular, how supervisor or organizational support for nurses can enhance resilience.

Similarly for social support, what actions of leadership at the unit and organizational level contribute to social support for nurses.

Reply: done

Regarding factors that contribute to resilience we add the following text in the introduction section.

…We should also recognize that several factors related to nurses’ work environment affect their resilience. In particular, literature suggests that increased levels of work engagement [29–31], perceived organizational support [32,33], and innovation-oriented nursing organizational culture [34] improve nurses’ resilience. Moreover, nurses that experience lower levels of workplace violence show higher levels of resilience [35]…

Regarding social support we add the following text in the introduction section.

…We should notice that a leadership that promotes a better nurses’ work environment may contribute to social support for them [43]. Moreover, organizational support may improve nurses’ social support [44]. Several other factors such as occupational commitment [45], sense of coherence [46,47], and positive strategies (i.e., occupational coping self-efficacy) [48] may contribute to nurses’ social support…

Comment

In the Methods section:

You present key elements of study design and describe in detail the setting, locations, and relevant dates, including periods of data collection. You report inclusion and exclusion criteria, and define all outcomes, predictors, and confounders.

For each variable of interest, you give sources of data and details of methods of assessment.

Moreover, you explain the calculation for the sample size, and describe all statistical methods, including those used to control for confounding.

Reply: done

Comment

In the Results section:

You report numbers of individuals at each stage of study, and give characteristics of study participants (e.g., demographic,) and information on predictors and potential confounders.

Also, you report unadjusted and adjusted estimates and their 95% confidence interval, and clarify which confounders were adjusted for and why they were included.

In my opinion, in the section on the demographic characteristics of the participants, there is no need for a detailed presentation of the results, as these are presented in Table 1.

Reply: done

We reduce the presentation demographic characteristics of the participants in results. We rewrite this part as follows.

Our sample consisted of 677 nurses with a mean age of 37.73 years. Among our nurses, 89.4% were females, and 54.4% possessed a MSc/PhD diploma. Among our nurses, 89.7% considered their health status as good/very good. Mean work experience was 11.59 years. Demographic characteristics of nurses are shown in Table 1.

Comment

In the Discussion section:

You summarise key results with reference to study objectives. Also, you give a cautious overall interpretation of results considering objectives, limitations, and results from similar studies. Moreover, you discuss limitations of the study, taking into account sources of potential bias or imprecision.

Two of the important findings of the study are that significant others support is negatively related to both anxiety and depression. As significant others include colleagues and supervisors, you should highlight their role in reducing nurses' anxiety and depression.

In addition, as you have measured various variables (e.g., resilience, anxiety, depression) in your study, please compare your finding with similar studies in Greece.

Reply: done

We add the following text in the Discussion.

Significant other support may include colleagues or/and supervisors for employees. Since nursing profession is a highly demand job with many challenges in work environment, nurses may consider that support from colleagues or/and supervisors is crucial. This is a possible explanation for our finding that significant others support was more important than family and friends support among our nurses. We should also recognize that may be exist mediators or confounders in the relationship between social support, anxiety, and depression that we did not measure in our study. For instance, personality characteristics and emotional intelligence may affect this relationship.

We add the following text in the Discussion.

…Moreover, our findings confirmed by a study in Greece including 380 nurses after the pandemic that found that 33.3% of nurses experience anxiety symptoms and 35.0% experience depressive symptoms [58]…

…We found that nurses in our sample experience moderate levels of resilience. These moderate levels of resilience were confirmed by two studies in Greece including nurses after the COVID-19 pandemic [59,60]. In particular, mean resilience score in our study was 3.50 while in these two studies ranged between 3.40 and 3.43…

We add three references from studies in Greece after the COVID-19 pandemic.

Reviewer 4 Report

Comments and Suggestions for Authors

I read the manuscript by Katsiroumpa et al entitled “Resilience and social support protect nurses from anxiety and depression: evidence from Greece in the post-COVID-19 era”. It is a very interesting topic, with a very satisfactory sample size. On the other hand, the manuscript presents very poor statistical analysis.

Here are my observations; I hope you find them useful.

Introduction

- I would ask the authors to refer to depressive symptoms and not depression in both the title and the rest of the manuscript. It should be clear to the reader that the manuscript is not about major depression, but about depressive symptomatology as recorded in questionnaires.

- At the end of the introduction, I would ask you to write in detail which research hypotheses you tried to confirm.

Materials and Methods

-For the MSPSS scale, please provide Cronbach's alpha for the subscales.

-Lines 150-152: I would suggest deleting the phrase, especially “…and, thus, we conducted linear regression analysis.” The central limit theorem, due to the sample size, covers the need for normality of the variables. I still doubt that all the continuous variables of this study would pass the Kolmogorov-Smirnov test (the linear regression analysis requires all variables to be multivariate normal).

-Since the authors raised the issue of assumptions, they should prove that the linear regression assumptions hold. In summary, I suggest that they check, multicollinearity, homoscedasticity, linearity, independence.

-Check the data for common method bias.

Results

-I believe it is necessary for the authors, in table 1, to give the scores of the variables also by gender. Please check for statistical differences by gender. Due to the sample size, it is also necessary to check with Hedges’ g.

-It is essential that authors provide a table with the correlations between continuous variables. The reader should be aware that there are not very high correlations between the variables. The addition of AVE (Average Variance Extacted) and CR (Composite Reliability) values ​​would be useful.

- The authors use anxiety and depression as independent variables. Thus, they create 2 models that do not include both variables at the same time. Can the authors support the independence of these two variables? Alternatively, I would suggest that they proceed with an extensive change to the Results. In the literature and in clinical practice, depressive symptomatology interacts with anxiety symptomatology.

Discussion

- I would ask the authors to provide more information on the situation of nurses in Greece regarding the variables examined in this manuscript. It would be of particular interest if the authors provided data on depression, anxiety and resilience in Greek nurses both during the pandemic crisis and before the pandemic. Compare your results with other studies on Greek nurses.

Author Response

Introduction

Comment

- I would ask the authors to refer to depressive symptoms and not depression in both the title and the rest of the manuscript. It should be clear to the reader that the manuscript is not about major depression, but about depressive symptomatology as recorded in questionnaires.

Reply: done

We replace “depression” with “depressive symptoms” in both the title and the rest of the manuscript.

Comment

- At the end of the introduction, I would ask you to write in detail which research hypotheses you tried to confirm.

Reply: done

We add the following text in the Introduction section.

In short, our research hypotheses were the following:

  • Resilience reduces anxiety in nurses.
  • Social support reduces anxiety in nurses.
  • Resilience reduces depressive symptoms in nurses.
  • Social support reduces depressive symptoms in nurses.

Materials and Methods

Comment

-For the MSPSS scale, please provide Cronbach's alpha for the subscales.

Reply: done

We add the following text in the Measurements section.

In particular, Cronbach’s alpha for the factor “family support” was 0.952, for the factor “friends support” was 0.954, and for the factor “significant others support” was 0.922.

Comment

-Lines 150-152: I would suggest deleting the phrase, especially “…and, thus, we conducted linear regression analysis.” The central limit theorem, due to the sample size, covers the need for normality of the variables. I still doubt that all the continuous variables of this study would pass the Kolmogorov-Smirnov test (the linear regression analysis requires all variables to be multivariate normal).

Reply: done

We delete the phrase, especially “…and, thus, we conducted linear regression analysis.”

Comment

-Since the authors raised the issue of assumptions, they should prove that the linear regression assumptions hold. In summary, I suggest that they check, multicollinearity, homoscedasticity, linearity, independence.

Reply: done

We check multicollinearity, multivariable normality, homoscedasticity, linearity.

We add the following text in the Statistical analysis section.

We assessed multicollinearity issues by calculating variance inflation factors (VIFs). VIF greater than 5 indicates multicollinearity between independent variables [53]. We examined histograms of the residuals to check multivariable normality.

Also, we add variance inflation factors in Tables 3 and 4. Please see Tables 3 and 4 in the manuscript.

Moreover, we add the following text in the results.

… There were no multicollinearity issues since VIFs for independent variables were lower than the acceptable value of 5. Figure 1 indicates multivariable normality since the residuals followed the normal distribution. Figure 2 indicates homoscedasticity and linearity of the multivariable model with anxiety as the dependent variable…

.. There were no multicollinearity issues since VIFs for independent variables were lower than the acceptable value of 5. Figure 3 indicates multivariable normality since the residuals followed the normal distribution. Figure 4 indicates homoscedasticity and linearity of the multivariable model with depressive symptoms as the dependent variable…

Moreover, we add Figures 1-4.

Dear Reviewer, we do not check for independence since our study design includes independent observations without repeated measurements. Thus, we consider that our data are independent a priori.

Comment

-Check the data for common method bias.

Reply: done

We add the following text in the Statistical analysis section.

We used the Harman's single factor to detect common method bias. In that case, common method bias is present when the unrotated solution produces one factor that accounts for more than 50% of variance [54,55]. We did not found common method bias in our study since our unrotated solution produced one factor that accounts for 38.1% of variance.

Results

Comment

-I believe it is necessary for the authors, in table 1, to give the scores of the variables also by gender. Please check for statistical differences by gender. Due to the sample size, it is also necessary to check with Hedges’ g.

Reply

Dear Reviewer, our apologies but we cannot understand this comment. We present demographic characteristics of nurses in Table 1. Table 1 includes, among, other, gender. Do you want to check for statistical differences between other demographics and gender in Table 1?

Please, take into your consideration our answer.

Comment

-It is essential that authors provide a table with the correlations between continuous variables. The reader should be aware that there are not very high correlations between the variables. The addition of AVE (Average Variance Extacted) and CR (Composite Reliability) values ​​would be useful.

Reply: done

We add correlations between variables. Please, see Table 3.

Dear Reviewer, we cannot understand how we can use AVE (Average Variance Extacted) and CR (Composite Reliability) values in this case. AVE and CR are useful in factor analysis. We did not perform factor analysis.

Comment

- The authors use anxiety and depression as independent variables. Thus, they create 2 models that do not include both variables at the same time. Can the authors support the independence of these two variables? Alternatively, I would suggest that they proceed with an extensive change to the Results. In the literature and in clinical practice, depressive symptomatology interacts with anxiety symptomatology.

Reply

Dear Reviewer, our apologies but we cannot understand this comment.

Probably, there is a misunderstanding. It is crystal clear even from the title of our manuscript (Resilience and social support protect nurses from anxiety and depression: evidence from Greece in the post-COVID-19 era) that we use anxiety and depression as dependent variables. Our independent variables were reliance and anxiety. We further explain this several times in our manuscript such as:

…In this context, our aim was to examine the impact of resilience and social support on nurses’ anxiety and depression…

…We constructed two multivariable linear regression models with anxiety and depression as the dependent variables…

We cannot see the reason to change our Results.

Please, take into your consideration our answer.

Discussion

Comment

- I would ask the authors to provide more information on the situation of nurses in Greece regarding the variables examined in this manuscript. It would be of particular interest if the authors provided data on depression, anxiety and resilience in Greek nurses both during the pandemic crisis and before the pandemic. Compare your results with other studies on Greek nurses.

Reply: done

We add the following text in the Discussion.

…Moreover, our findings confirmed by a study in Greece including 380 nurses after the pandemic that found that 33.3% of nurses experience anxiety symptoms and 35.0% experience depressive symptoms [73]…

…We found that nurses in our sample experience moderate levels of resilience. These moderate levels of resilience were confirmed by two studies in Greece including nurses after the COVID-19 pandemic [74,75]. In particular, mean resilience score in our study was 3.50 while in these two studies ranged between 3.40 and 3.43…

Round 2

Reviewer 1 Report

Comments and Suggestions for Authors

The authors have adequately addressed the previous comments and revised the manuscript accordingly. The responses provided are satisfactory, and the revisions improve the clarity and accuracy of the manuscript.

However, one minor correction remains. The term "coefficient beta" should be revised to "beta coefficient" (for instance, "adjusted standardized coefficient beta" should be "adjusted standardized beta coefficient") throughout the abstract, main text, and tables to align with standard statistical conventions.

Author Response

Reviewer: 1

Dear Reviewer,

Thank you very much for the peer review of the manuscript “Resilience and social support protect nurses from anxiety and depression: evidence from Greece in the post-COVID-19 era.". Thank you for your comment, which has improved the quality of the manuscript. We have addressed the comment in the revised text.

Please find below the answer to your comment. Hoping the revised manuscript fulfills the journal’s standards, we thank you for your courtesy.

We are looking forward to your response.

Yours sincerely,

Petros Galanis, Assistant Professor

Comment

However, one minor correction remains. The term "coefficient beta" should be revised to "beta coefficient" (for instance, "adjusted standardized coefficient beta" should be "adjusted standardized beta coefficient") throughout the abstract, main text, and tables to align with standard statistical conventions.

Reply: Done

We replaced the term “adjusted standardized coefficient beta” with the term “adjusted standardized beta coefficient” throughout the manuscript.